# COMPETITION PRIORS FOR OBJECT-CENTRIC LEARNING

## ABSTRACT

Humans are very good at abstracting from data and constructing concepts that are then reused. This is missing in current learning systems. The field of object-centric learning tries to bridge this gap by learning abstract representations, often called slots, from data without human supervision. Different methods have been proposed to tackle this task for images, whereas most are overly complex, non-differentiable, or poorly scalable. In this paper, we introduce a conceptually simple, fully-differentiable, non-iterative, and scalable method called **COP** (**C**ompetition **O**ver **P**ixel features). It is implementable using only Convolution and MaxPool layers and an Attention layer. Our method encodes the input image with a Convolutional Neural Network and then uses a branch of alternating Convolution and MaxPool layers to create competition and extract primitive slots. These primitive slots are then used as queries for a variant of Cross-Attention over the encoded image. Despite its simplicity, our method is competitive or outperforms previous methods on standard benchmarks. The code is publicly available.

## 1 INTRODUCTION

Humans excel at abstracting from raw input to create useful symbolic-like representations for objects, which can then be applied, reused, and combined for reasoning or decision-making (Spelke & Kinzler, 2007). However, despite the rise of Deep Learning (Krizhevsky et al., 2012; Schmidhuber, 2015) and the access to large scale compute and data (Deng et al., 2009) that led to human level performance in various domains (He et al., 2016; Mnih et al., 2015; Silver et al., 2016), neural networks do not posses this abstraction ability.

Various recent works tried to address the abstraction problem by learning abstract object-centric representations — often known as slots (Locatello et al., 2020; Wu et al., 2023; Chang et al., 2023; Engelcke et al., 2019; Burgess et al., 2019) — from high-dimensional input, such as images, videos, 3D point clouds or even 3D scenes (images of different angles onto the same objects). However, most of the currently proposed methods are complex and partially suffer from training instabilities (Chang et al., 2023). Many of these methods are derived from the original Slot Attention idea (Locatello et al., 2020), which applies a modification of Cross-Attention (Vaswani et al., 2017) in an iterative refinement procedure. We argue that such an iterative refinement procedure is not desired and propose a solution that can serve as a simple baseline.

Prior to explicit slot extraction methods, *competition mechanisms* (Srivastava et al., 2013) like Max-Pooling or Spatial Pyramid MaxPooling (He et al., 2014) showed a strong performance (He et al., 2015; 2017; Szegedy et al., 2014) to obtain highly relevant abstract features. Srivastava et al. (2014b) empirically showed that competition among subnetworks leads to specialization, whereas for different input, different subnetworks are activate. We argue that this ability to specialize and abstract are useful properties for learning object-centric representations.

In this paper, we present a novel method that we call Competition Over Pixel features (COP). COP is a simple, scalable, non-iterative and fully-differentiable approach to extract slots from images with simple Convolutions (CNN; LeCun et al., 2004) and MaxPool blocks (Krizhevsky et al., 2012) that are combined with a Soft-Winner-Take-All attention (SWTA-Attention) layer (inspired by Cross-Attention Vaswani et al., 2017). The effectiveness of COP demonstrates that the iterative nature of Slot Attention based methods is not necessary.

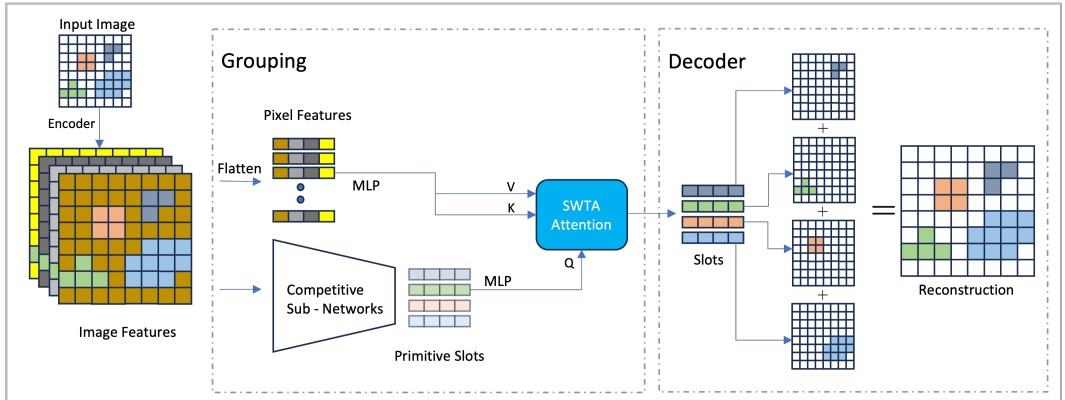

Figure 1: *Grouping:* We learn Primitive Slots from image features using Competitive Sub-Networks. We obtain pixel features by flattening all the image features from the encoder. We pass pixel features and Primitive Slots to a SWTA layer, where *Keys* (K) and *Values* (V) are the pixel features and *Queries* are the Primitive Slots. SWTA-attention outputs the slots and is similar to Cross-Attention, except we take a softmax over the queries, instead of the keys. The decoder is applied on every slot separately to reconstruct the input and an mask. A softmax is applied to the masks along the pixel dimension (for simplicity the masks are not shown in the figure). The final reconstruction is obtained by performing a weighted sum of all the individual reconstructions across the pixels with the weights coming from the masks.

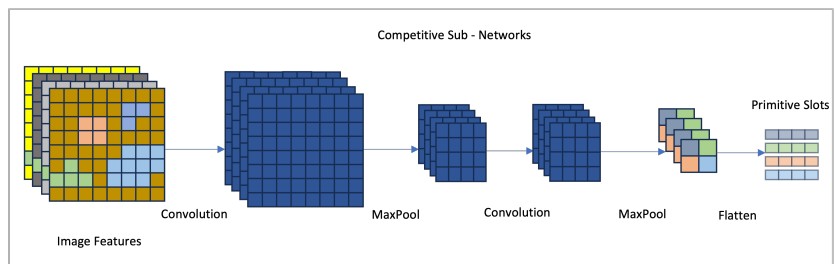

Figure 2: *Competitive Sub-Networks:* We use alternating Convolution and MaxPool layers. After these layers, we flatten features to obtain Primitive Slots. The architecture along with the slot-wise reconstruction in the decoder, induces competition between sub-networks. The sub-networks are forced to explain different parts of the input. Therefore, the resultant Primitive Slots are good queries for the SWTA-Attention layer.

Our main contributions are:

- We propose COP (Competition Over Pixel features) a novel method for learning object-centric representations.
- COP is a simple baseline for abstraction models, since it is non-iterative and consists of vanilla building blocks like CNN, MaxPool layers and a modified Cross-Attention.
- We evaluate COP on standard Object-Centric benchmarks, where it is competitive or out-performs various other slot attention methods.

## 2  RELATED WORK

Object-centric learning is related to abstraction, which is a well studied topic in different areas of Machine Learning (Givan et al., 2003; Ravindran & Barto, 2003; Sutton et al., 1999; Li et al., 2006; Vezhnevets et al., 2017; Kulkarni et al., 2016; Patil et al., 2022). Object-centric learning methods try to extract representations of objects given a input, which could be images or some other modality. These extracted representations are known as slots Locatello et al. (2020).

Many such methods consist of three modules: an image encoder, a grouping module and a decoder. The image encoders are typically realized with multiple CNN layers (Locatello et al., 2020; Engelcke et al., 2021). For the decoder, the spatial broadcast decoder (Watters et al., 2019) — or slight deviations thereof — is the de-facto standard, since it provides a good inductive bias for learning disentangled representations (Engelcke et al., 2020). The main difference between common slot extraction methods is the *grouping module*. Most grouping modules can be divided into (a) graph-based, (b) generative and (c) iterative refinement based approaches.

**Graph-based approaches**   Pervez et al. (2022) use a graph-cutting approach to perform clustering of pixel-features via a quadratic program. The disadvantage of this approach is that solving the underlying quadratic program is computationally intense and poorly parallelizable.

**Generative approaches**   aim to learn parameterized distributions of latent variables by minimizing a proxy of the log-likelihood. Works such as IODINE (Greff et al., 2019), MONet (Burgess et al., 2019), SPACE (Lin et al., 2020) or Genesis (Engelcke et al., 2019) fall into this category. While these methods are theoretically grounded, they lack empirical performance. Genesis-V2 Engelcke et al. (2021) is an extension of Genesis and one of the first fully-differentiable methods that is able to choose the number of slots *by design* during training and inference (i.e., not by doing a second forward pass with a different number of slots). Although this selection option brings some interesting properties, the method has not found wide adoption. We argue that the reason for this is that it is complex in terms of math and code.

**Iterative refinement approaches**   The Slot Attention module (Locatello et al., 2020) — often simply called Slot Attention —is a successful and widely adapted technique. Thus, there exists a sizable body of work that is dedicated to improve it: Implicit Slot Attention (ISA Chang et al., 2023) stabilizes the iterative refinement process of Slot Attention by learning the fixed-point of the iterative refinement procedure with a first-order Neumann series approximation. While it stabilizes the training, it tends to need even more iterations than Slot Attention which increases the overall time complexity. Jia et al. (2023) extend Implicit Slot Attention by learning distributions for initializations of the iterative refinement procedure. An other improvement suggestion to the original Slot Attention module came from (Kim et al., 2023) who introduced a locality-bias as it is common for CNNs in the attention part of Slot Attention. Similarly, Gao et al. (2023) improved the slot initialization with explicit clustering methods, such as Mean-Shift (Comaniciu & Meer, 2002) or K-means (Lloyd, 1982). SLATE (Singh et al., 2022a) uses Slot Attention as the grouping module, but replaces the CNN encoder with a discrete Variational Autoencoder (Ramesh et al., 2021) encoder and the CNN spatial broadcast decoder (Watters et al., 2019) with an autoregressive transformer decoder (Vaswani et al., 2017).

**Applications**   In this work we present a slot extraction method. We therefore point the interested reader to recent applications, including object-centric learning on videos (Singh et al., 2022b; Elsayed et al., 2022; Tang et al., 2023), point clouds (Wang et al., 2021), neural scene rendering (Sajjadi et al., 2022b;a) inspired by Mildenhall et al. (2020) or reinforcement-learning (Veerapaneni et al., 2019).

**Competition in Neural Networks**   Competition in artificial neural networks is inspired by observations of the human brain. Competitive dynamics among neurons and neural circuits have played a crucial role in understanding brain processes in biological models Eccles et al. (1967); Ermentrout (1992). The significance stems from early findings that revealed a recurring "on-center, off-surround" neural architecture in various brain regions Ellias & Grossberg (1975). The "on-center, off-surround" architecture involves neurons providing excitatory feedback locally and inhibitory signals broadly. Therefore it creates competition between sub-networks (Andersen et al., 1969; Ellias & Grossberg, 1975). The biological model in Lee et al. (1999) is also of interest, as it proposes that attention activates a winner-take-all competition over the visual features in the human brain. Inspired from the brain, such competitive networks have made there way to machine learning (Maass, 1999; 2000). Hamming Networks (Lippmann, 1987) perform iterative lateral inhibition between competitors. Local-Winner-Take-All networks (Srivastava et al., 2014a) setup blocks of neurons, and pass the winning activation of each block to the next layer. Adding sources of competition on networks

trained on faces, showed impressive abstraction capabilities, where feature maps of deeper layers (i.e., closer to the output) become more abstract and depict parts of a face (Wang & Tan, 2014).

# 3 METHOD

We propose *Competition Over Pixel features*, a method to learn Object-Centric representations from images. Most architectures consist of three main modules: an encoder, a grouping and a decoder module. The encoder computes useful features from the raw image, the grouping module extracts object-representations (slots) and the decoder reconstructs the individual slots to partial images, which are then summed up to obtain the input image in an auto-encoder fashion. The main novelty of COP lies in its grouping mechanism (Figure 1 and 2). COP groups image features using competitive sub-networks and a variant of the attention layer. Competition is induced in COP due to the following components: 1) Winner Take All MaxPool layers 2) SWTA-Attention 3) Spatial Broadcast Decoder.

In the following sub-section (3.1), we first go through the architecture of COP. Afterwards, we describe the sources of competition in COP (sub-section 3.2).

## 3.1 ARCHITECTURE

The COP architecture consists of an encoder, grouping module and a decoder.

**Encoder** The encoder is a simple image encoder with CNN layers LeCun et al. (2004). We have a fixed kernel size, padding and stride across these layers. The layers are setup in such a way that the spatial dimension of the input is preserved. For example, if the input image is of size $H \times W$ then the encoder output is $H \times W \times c$, where, $H$ and $W$ are the height and width of the image. $c$ is the number of filters used in the last layer. This is done to obtain features at pixel level, which can then be later grouped together to obtain slots. Similar to Locatello et al. (2020), we also augment the pixel features with a positional embedding. For more details, see Appendix A.2.

**Grouping** The output of the encoder is fed to COP's grouping module, which outputs slots. The grouping module consists of competitive sub-networks and a SWTA-Attention.

The competitive sub-networks ingest the encoder output and return primitive-slots. A competitive sub-networks consists of MaxPool and Convolution layers. Similar to the encoder, we build the CNN layers in such a way that they preserve the spatial dimension. But, the MaxPool layers reduce it (Figure 2). After applying CNN and MaxPool layers we obtain an output of size $n_h \times n_w \times c$. Here, $n_h, n_w$ are height and width of the output and $c$ is the number of filters of the last CNN layer. We flatten the output to obtain, $n \times c$, where $n = n_h \times n_w$ are the number of primitive slots. Conceptually, $n$ is therefore a hyperparameter of the network. We pass these primitive-slots through an MLP layer and then use them as queries in the SWTA-Attention.

The SWTA-Attention is a variant of the standard attention. Instead of taking a softmax across the keys, it takes a softmax over the queries. This creates competition in the slots to explain different parts of the input. Something similar is also also done in the Slot Attention Module (Locatello et al., 2020) to create competition between slots. However, in their case, they take a softmax over the keys and then normalize the attention weights over the queries. Another key difference is that they apply it in an iterative refinement fashion, while we apply it only once. The attention layer can also be thought of as a storage of patterns (Ramsauer et al., 2020; Widrich et al., 2021; Paischer et al., 2022), from which we retrieve patterns similar to primitive slots.

COP first flattens the encoder output and passes it through MLPs so they can be used as keys ($\mathbf{K} \in \mathbb{R}^{P \times D}$) and values ($\mathbf{V} \in \mathbb{R}^{P \times D}$) in the SWTA-Attention layer, whereas $P$ is the number of pixels and $D$ is the dimensionality of the slots. The primitive-slots are used as queries ($\mathbf{Q} \in \mathbb{R}^{n \times D}$). Once we get the attention coefficients, we can use them to get the slots ($\mathbf{S} \in \mathbb{R}^{n \times D}$) as follows by using

$$\mathbf{S} = \mathbf{W}^T \mathbf{V} \text{ where, } \mathbf{W} = \text{softmax}(\frac{\mathbf{K}\mathbf{Q}^T}{\tau}). \tag{1}$$

The temperature $\tau$ is set to $\sqrt{n}$, whereas $n$ is the number of slots. This is a lower temperature then commonly used in Cross-Attention (Vaswani et al., 2017) but the same as used in Locatello et al.

(2020). Empirically we found that the resulting lower-entropy distribution of attention values leads to better results.

**Decoder** The decoding treats each slot separately with a Spatial Broadcast Decoder (Greff et al., 2019). Every slot reconstructs its own image and a mask. The masks are normalized and a softmax along the pixel dimension of all masks mixes all the images together. We train the model end to end with a mean squared error reconstruction loss (similar to Greff et al., 2019; Locatello et al., 2020).

Our method is able to extract slots, without any iterative refinement. In the next subsection, we explain how COP creates competition amongst the slots to explain different parts of the input.

## 3.2 COMPETITION IN COP

COP creates competition through different sources: The MaxPool Layers, the SWTA-Attention layer and a slot wise reconstruction decoder.

**Competition through MaxPool layers** MaxPool layers are interleaved between Convolution layers (Figure 2). MaxPool layers work by transmitting activations of winning units to the next layer. As there is only one winner for a local group of neurons, the layer activations are sub-sampled. This results in sub-networks competing to have higher activations. During back-propagation, units that win and the subnetworks that are responsible for this will get updated. As a result, a winning sub-network is reinforced to win more if it predicts correctly. If a sub-network wins, and does not predict correctly, it is not reinforced. One can also look at this as a gradient-based search over finding sub-networks, which explain the input correctly. An important side effect of these competitive sub-networks is that the resulting primitive-slots explain different parts of the input. A neuron has a higher chance of winning if it explains a different part of the input, rather than explaining the same feature as another neuron.

**Competition through SWTA-Attention** Another source of competition for COP is the SWTA-Attention layer. The layer takes a softmax across queries, this forces the queries to compete for keys (pixel features). We see improvement in results by using this layer, over not using it (See Table 3).

**Competition through Spatial Broadcast Decoder** Finally, we use a slot wise reconstruction decoder, i.e., every slot is separately fed to the decoder for reconstruction (Figure 1). The final reconstruction is obtained by mixing all reconstructions with a softmax over the respective masks. This creates competition over pixels in the final reconstructed image. Due to the mixing of the reconstructions, the slots get reinforced for explaining different parts of the input. Thus, providing a push to the competitive nature of the slots. This has further been discussed in Engelcke et al. (2020).

In summary, MaxPool layers, SWTA-Attention layer and the decoder lead to competition amongst the slots and as a result they try to explain different parts of the input.

## 3.3 ADVANTAGES

The main advantage of our method over Slot Attention is, that it is non-iterative and therefore provides a better time and space complexity (see table 1).

Table 1: Runtime and Memory Complexities of the grouping module of Slot Attention vs. COP. $S$ denotes the number of slots, $D$ is the dimensionality of the slots and the number of channels (these are assumed to be equal for simplicity), $H$ and $W$ are the height and width of the image. While COP uses another few layers for the competition, Slot Attention performs several iterations of the iterative refinement procedure.

|  | **Slot Attention** | **COP (ours)** |
|---|---|---|
| Time (Training) | $\mathcal{O}(T \cdot S \cdot H \cdot W \cdot D)$ | $\mathcal{O}((L_{\text{comp}} + S) \cdot H \cdot W \cdot D)$ |
| Space (Training) | $\mathcal{O}(T \cdot S \cdot H \cdot W \cdot D)$ | $\mathcal{O}((L_{\text{comp}} + S) \cdot H \cdot W \cdot D)$ |
| Time (Test) | $\mathcal{O}(T \cdot S \cdot H \cdot W \cdot D)$ | $\mathcal{O}((L_{\text{comp}} + S) \cdot H \cdot W \cdot D)$ |
| Space (Test) | $\mathcal{O}(S \cdot H \cdot W \cdot D)$ | $\mathcal{O}((L_{\text{comp}} + S) \cdot H \cdot W \cdot D)$ |

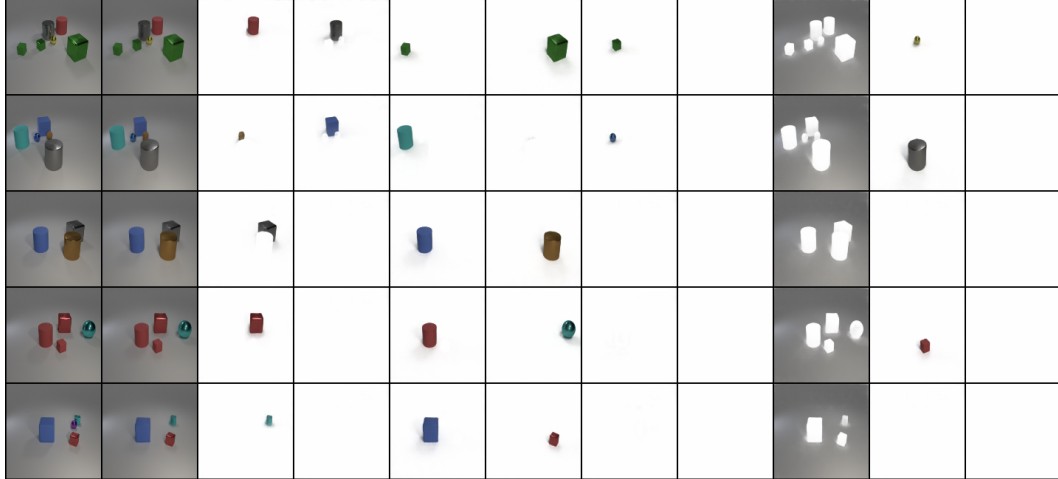

Figure 3: *Results on CLEVR6:* Visualized results of Multi-dSprites. The first column is the original image. The second column is the final reconstruction by the model, namely the weighted sum of individual reconstructions. Columns 3-11 are reconstructions of individual slots. The individual reconstructions are displayed without the mask.

**Scalability** The Slot Attention module is an **iterative refinement method** that needs several iterations to converge to a fixed point. As the complexity of the task increases, so does the number of required iterations. Chang et al. (2023), while addressing some of the training instabilities of Slot Attention, require even more iterations. In the case of training on CLEVR, Chang et al. (2023) may utilize up to 11 iterations. These iterative methods, however, pose a significant challenge when it comes to scaling them for large-scale datasets.

COP distinguishes itself by not relying on recurrent neural networks (RNNs) and being a non-iterative method. This characteristic makes COP particularly well-suited for scaling up to handle large datasets and computational demands efficiently. COP's non-iterative nature simplifies its application to extensive data processing and computation tasks, providing a distinct advantage over iterative methods like Locatello et al. (2020) and Chang et al. (2023).

## 4 EXPERIMENTS

**Baselines** We compare COP with the following commonly used slot extraction methods: Slot Attention (Locatello et al., 2020), IODINE (Greff et al., 2019) and MoNET (Burgess et al., 2019).

**Datasets** For the experiments we use three standard benchmarks common in prior work, namely CLEVR6, Multi-dSprites and Tetrominoes (introduced in Kabra et al., 2019). These are synthetic datasets which provide ground truth segmentation masks for evaluation. CLEVR consists of rendered scenes of 3D geometric shapes, e.g., spheres, cubes or cuboids, with a single light source. CLEVR6 is a subset of CLEVR which only contains a maximum of six objects. Multi-dSprites consists of 2-5 objects, which are 2D shapes (ellipses, hearts, squares) in different colors on black background. Tetrominoes consists of always three objects, namely 2D shapes from the tetris game.

Occlusion is an important property that largely defines the difficulty of a dataset. In tetrominoes there is no occlusion between objects. For CLEVR6, objects may partially occlude each other, though usually a single object is not occluded by more than one other object. In Multi-dSprites an object may be occluded by several other objects, which makes it the most challenging of the three tasks in this regard.

For all these datasets we follow the protocol mentioned in Slot Attention (Locatello et al., 2020). CLEVR6 consists of 70k, and Multi-dSprites and Tetrominoes of 60k training samples. We partition the training set into a training and validation split to choose hyperparameters. For testing we trained on the whole training set and tested on 320 test samples (same as Slot Attention).

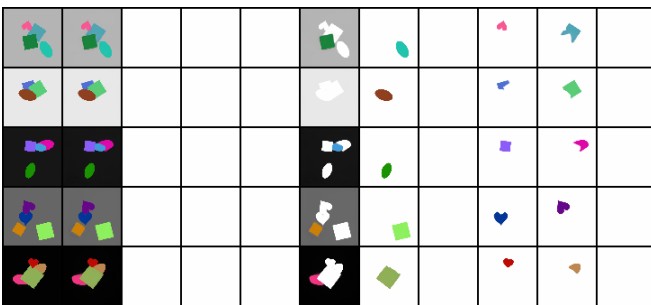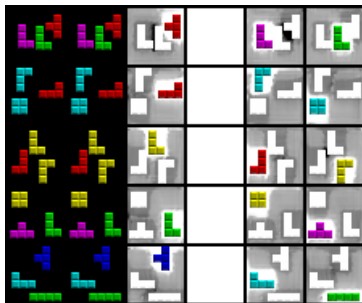

Figure 4: *Left:* Reconstructions of Multi-dSprites. The first column is the original image. The second column is the weighted sum of individual reconstructions where the weights come from the masks to which a pixel-wise softmax was applied. Columns 3-11 are reconstructions of individual slots. *Right:* Reconstructions of Tetrominoes. Again, the first column is the original image, while the second column is the weighted sum of individual reconstructions. Columns 3-6 are reconstructions of individual slots. The individual reconstructions are displayed without the mask.

**Training** For the encoder and decoder architectures we follow the protocol described in (Locatello et al., 2020). For the grouping module we use the architecture described in section 3 with slight modifications for each dataset. First, we adjust the number of slots to equal or somewhat more than maximum number of objects of the according dataset. Since we have seen degrading performance of too many slots (also see section 5). Specifically, for CLEVR6 and Multi-dSprites we use nine slots, and for Tetrominoes we use four slots (three object slots and one background slots). All datasets are trained using a mean-squared-error reconstruction loss. Full details about the hyperparameters can be found in the appendix A.2.

**Evaluation metric** All of the above-mentioned datasets contain ground truth masks that enable evaluation of the image segmentations. We use the Adjusted Rand Index (Rand, 1971; Hubert & Arabie, 1985) as evaluation metric. This is a metric in the interval $[-0.5, 1.0]$ that compares two cluster assignments, whereas a score of $1$ means the clusters are identical, a score of $0$ would mean roughly equal to random assignment and $-0.5$ the worst assignment possible (worse than random). As common in the literature (Locatello et al., 2020; Engelcke et al., 2021) we omit the background ground truth masks in the evaluation, since we mostly want to distinguish between foreground objects. This metric is referred to as the FG-ARI.

**Results** COP is competitive for CLEVR6, but outperforms Slot Attention in Multi-dSprites and Tetrominoes. COP achieves state-of-the-art performance — on par with Slot Attention — when averaged over all three datasets. An overview of the results is given in table 2. Qualitatively we can see in figures 4 and 3 that the slot reconstructions are semantically meaningful and of high quality. Interestingly, some slots seem to specialize. E.g., for CLEVR6 and Tetrominoes there seems to be a clear specialisation of the second slot to capture the gray background. This is also consistent with our observations of the slots over the training process. Non-background slots don't seem to specialize in the position, shape, size nor color of the objects. Interestingly, some slots are not used in *any* of the reconstructions, which can be regarded as a specialisation on "not interfering" with other reconstructions.

Table 2: Results of different slot extraction methods on several datasets. Prior methods averaged over five seeds, while our results were averaged over three seeds.

| Method | CLEVR6 | Multi-dSprites | Tetrominoes | Average |
|---|---|---|---|---|
| Slot Attention | **98.8** $\pm 0.3$ | $91.3 \pm 0.3$ | $99.5 \pm 0.2$ | **96.53** |
| IODINE | **98.8** $\pm 0.0$ | $76.7 \pm 5.6$ | $99.2 \pm 0.4$ | 91.57 |
| MONET | $96.2 \pm 0.6$ | $90.4 \pm 0.8$ | n/a | 93.30 |
| **COP (ours)** | $97.6 \pm 0.6$ | **92.3** $\pm 0.2$ | **99.8** $\pm 0.1$ | **96.57** |

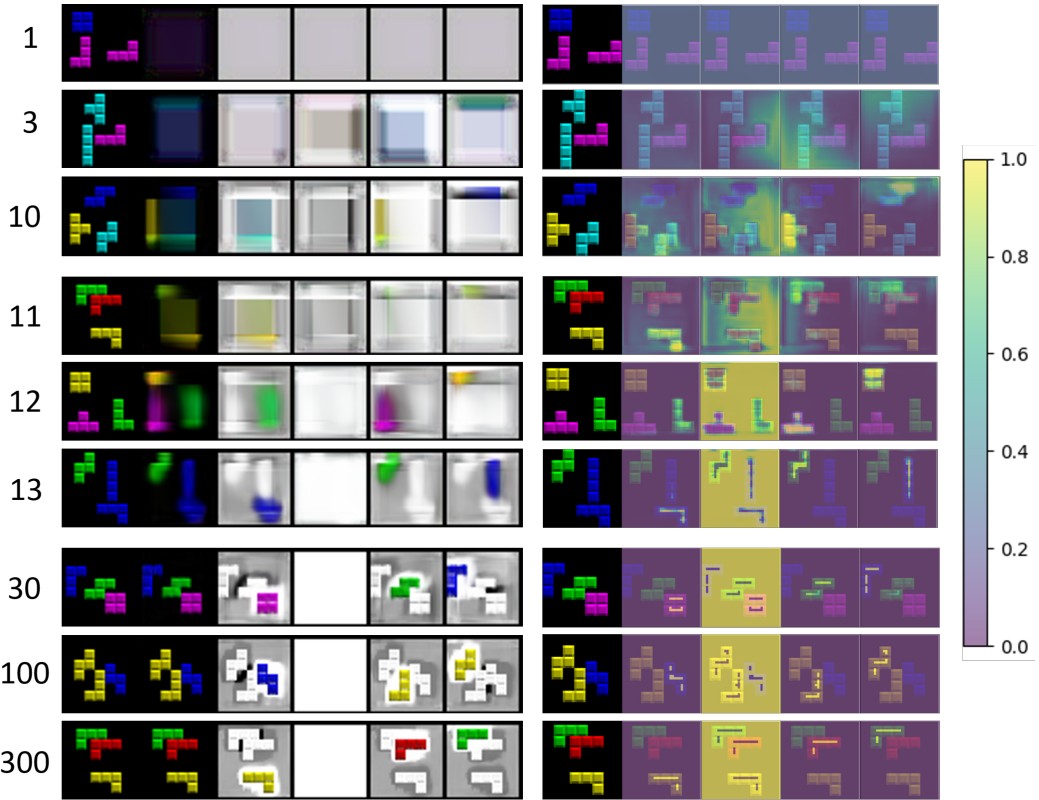

Figure 5: *Reconstructions and visualized attention heatmaps of slots over pixel features during training on Tetrominoes:* The numbers on the left denote the completed training epochs, the left group of images are reconstructions, whereas the right group are visualized attention maps. The columns of the reconstruction images are in the following order: (col. 1) ground truth image, (col. 2) final reconstruction (cols. 3-6) individual slot reconstructions. The columns of the visualized attention maps are in the following order: (col.1) ground truth image, (cols. 2-5) individual attention maps of the queries over the keys (i.e. the projected pixel features).

## 5 ANALYSIS AND ABLATION STUDIES

**Learned query visualization** We visualized the attention of a slot over the keys of the pixel features. The results on the Tetrominoes dataset are in figure 5 (for further visualizations, please see appendix A.1). For the Tetrominoes run in figure 5, we can roughly identify three phases.

- **Warmup phase** Up until epoch 10, the reconstructions are poor and the attention maps are not sharp nor object-specific.
- **Transition phase** From epoch 11 to 13, there is a transition phase. While reconstructions are still poor after epoch 11, they have become sharper and object-specific after epoch 13. Also the attention maps clearly indicate that the queries are focusing on object-specific parts of the input.
- **Refinement phase** The objects at epoch 30 are still somewhat blurry, but are more detailed refined over the next several hundred epochs. For the attention maps, we can see that between epoch 13 and epoch 30 the attention maps change little. From epoch 30 onwards, the queries attend to very similar pixels of individual objects.

**Which attention mechanism to use** To understand the impact of the attention mechanism, we tried several ablations.

- **COP w/o SWTA-Attention** To check if SWTA-Attention over pixel features is beneficial at all, we omit the attention mechanism and directly reconstruct the images from the queries.

- **COP w/ Cross-Attention** Furthermore we substitute the SWTA-Attention with the common Cross-Attention (Vaswani et al., 2017).

- **COP w/ Attention as in Slot Attention** we substitute the SWTA-Attention with the attention variant used in Slot Attention (Locatello et al., 2020). Note that the GRU, MLPs at the end of each iteration were not used. Also we just applied the attention-variant once in a non-iterative fashion.

The results can be found in table 3. With four slots not using an attention-variant leads to a substantial drop in the FG-ARI score by 1.8. We conclude that the introduced competition of slots over pixel features is a useful inductive bias for this architecture.

**The effect of the maximum number of slots** To gain further insights on how COP generalizes to more slots, we ran an ablation on Tetrominoes (Table 3). While COP manages to efficiently solve the task for six slots — which are 50% more than the ideal number slots — the performance largely degrades at nine or more slots. As visible in figure 8, oftentimes information about a single object is then shared among two slots.

Table 3: FG-ARI Mean and standard deviations over three seeds on the Tetrominoes dataset. *Left:* Ablations on the number of maximum slots. The ideal number of slots is four (i.e., three foreground objects and the background). If the number of slots is six (i.e., 50% higher than the ideal number of slots), the model still learns reasonably. If the number of slots becomes twice the number of ideal slots, the models performance degrades rapidly. *Right:* A comparison of attention-variants. The currently chosen attention-variant has the strongest performance. Using no attention mechanism (i.e., using the queries directly for reconstruction), Cross-Attention or the attention mechanism used in the Slot Attention module degrade performance.

| Configuration | Tetrominoes | Attention variants (4 slots) | Tetrominoes |
|---|---|---|---|
| **COP (4 slots)** | **$99.77 \pm 0.12$** | **COP** | **$99.77 \pm 0.12$** |
| COP (6 slots) | $99.58 \pm 0.65$ | COP w/o Attention | $97.97 \pm 0.20$ |
| COP (8 slots) | $89.13 \pm 18.58$ | COP w/ Attn as in Slot Attention | $91.07 \pm 15.23$ |
| COP (9 slots) | $84.21 \pm 13.62$ | COP w/ Cross-Attention | $90.92 \pm 14.87$ |
| COP (16 slots) | $63.99 \pm 1.57$ | | |

## 6 LIMITATIONS

**Sensitivity to the maximum number of slots** One down-side of COP is, that the number of objects is not adjustable during or after training. Other methods also suffer from a sensitivity to the number of slots during training. This problem can be improved by training with more slots and flexibly adjust the number of slots during inference (Zimmermann et al., 2023).

## 7 CONCLUSION

In this paper we presented a novel method for object-centric learning that is based on competition, called Competition Over Pixel features (COP). COP is simple, scalable, fully-differentiable and contrary to the widely applied Slot Attention, it is non-iterative. It comes with the down-side of not being able to adjust to a different number of slots between training and test time. Empirically our method is competitive or outperforms existing slot extraction methods.

## 8 REPRODUCIBILITY STATEMENT

The code for reproducing our results is publicly available. Detailed information on the hyperparameters can be found in the appendix A.2.

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

## A  APPENDIX

### A.1  ABLATIONS

In this section we add additional figures for the ablation studies. The figures are taken from **the best prediction of the worst run** among all the seeds. Figures 6, 7 and 8 are reconstructions of the respective architectures.

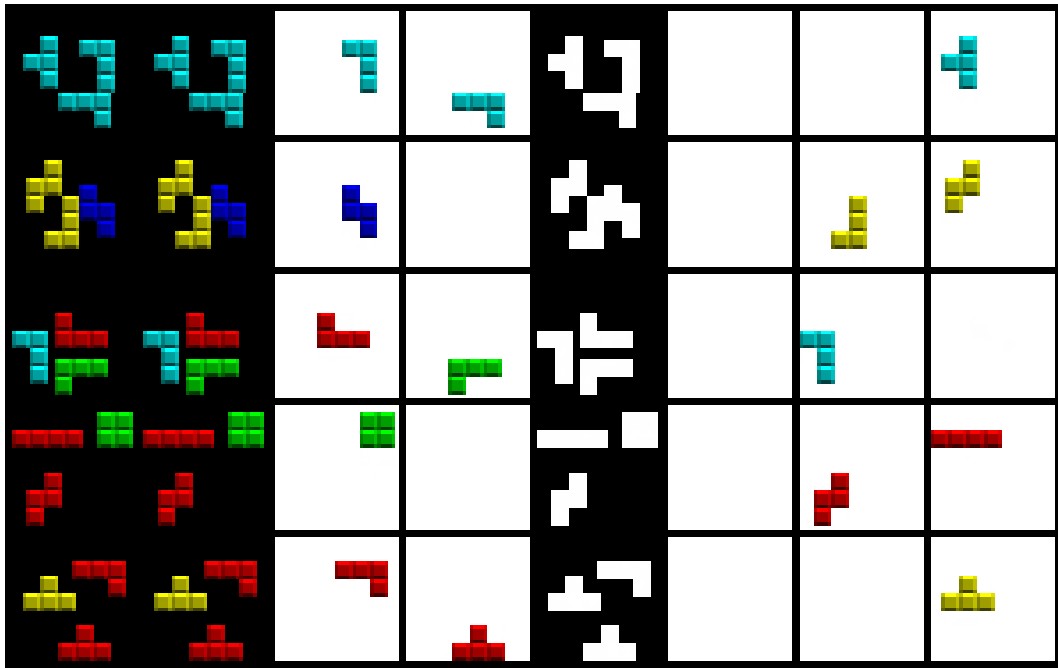

Figure 6: *Ablation of Tetrominoes with six slots:* The reconstructions still work reasonably well.

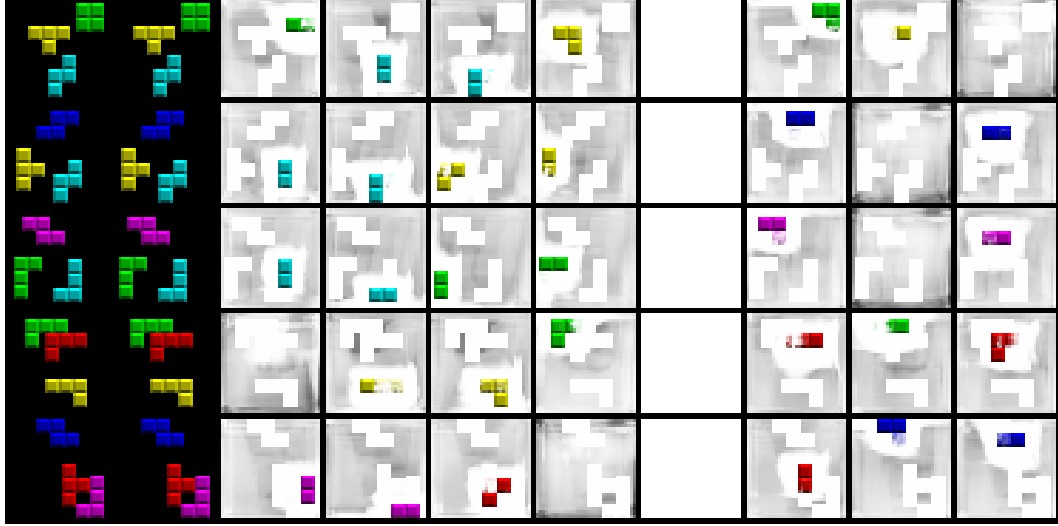

Figure 7: *Ablation of Tetrominoes with 8 slots:* Since there are twice as many slots than the ideal four for Tetrominoes (i.e., three objects, one background), a single object is often represented in two slots.

## A.2 HYPERPARAMETERS

### A.2.1 ENCODER ARCHITECTURES

For the encoder architectures we follow the protocol given in Locatello et al. (2020). The detailed layer descriptions for CLEVR6 can be found in table 4. Similarly the encoder layer descriptions for Multi-dSprites and Tetrominoes are explained in table 5.

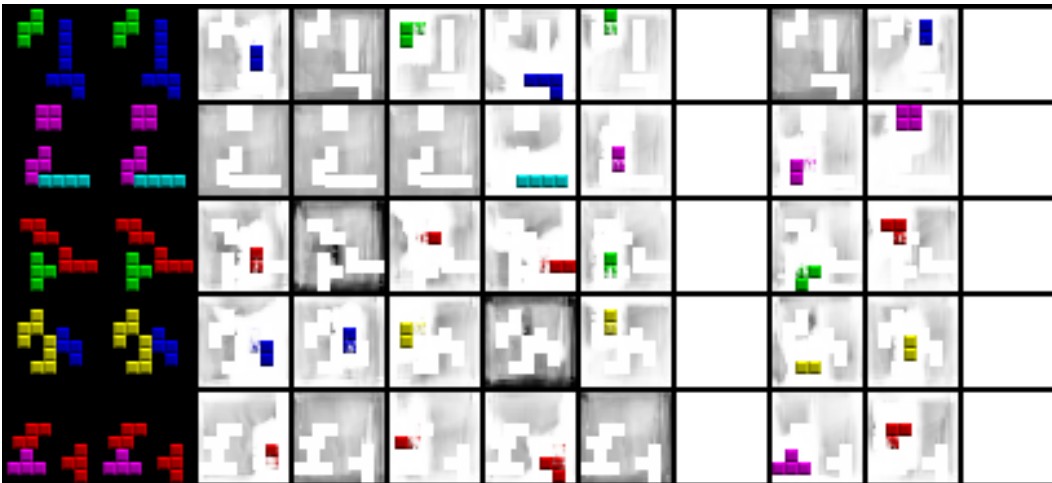

Figure 8: *Ablation of Tetrominoes with nine slots:* Since there are more than twice as many slots than the ideal four for Tetrominoes (i.e., three objects, one background), a single object is often represented in two slots.

Table 4: CNN encoder for CLEVR6.

| Type | Size/Channels | Activation | Comment |
|---|---|---|---|
| Conv $5 \times 5$ | 64 | ReLU | Stride: 1 |
| Conv $5 \times 5$ | 64 | ReLU | Stride: 1 |
| Conv $5 \times 5$ | 64 | ReLU | Stride: 1 |
| Conv $5 \times 5$ | 64 | ReLU | Stride: 1 |
| Position Embedding | - | - | See Section A.2.4 |
| Flatten | axis: $[0, 1 \times 2, 3]$ | - | Flatten x, y pos. |
| Layer Norm | - | - | - |
| MLP (per location) | 64 | ReLU | - |
| MLP (per location) | 64 | - | - |

### A.2.2 COMPETITION ENCODER ARCHITECTURES

The layer descriptions for the competition encoders can be found in tables 6, 8 and 7. Note that before the Competition encoder, a positional bias is added to the pixel features, as described in appendix A.2.4.

### A.2.3 DECODER ARCHITECTURES

Again, we follow the protocol of Locatello et al. (2020). Note that before the decoding, a positional bias is added to the spatial map, as described in appendix A.2.4. The detailed layer descriptions for CLEVR6 can be found in table 9. Similarly the encoder layer descriptions for Multi-dSprites and Tetrominoes are explained in table 10.

### A.2.4 POSITIONAL EMBEDDING

We follow the positional embedding given in Locatello et al. (2020). Summarized, for each pixel a 4-dimensional positional embedding is initialized that encodes the distance to all colors (normalized to the range $[0, 1]$). These 4 dimensional embeddings are then projected with a learnable matrix to the dimensionality of the pixel features.

Table 5: CNN encoder for Multi-dSprites and Tetrominoes.

| Type | Size/Channels | Activation | Comment |
|---|---|---|---|
| Conv $5 \times 5$ | 32 | ReLU | Stride: 1 |
| Conv $5 \times 5$ | 32 | ReLU | Stride: 1 |
| Conv $5 \times 5$ | 32 | ReLU | Stride: 1 |
| Conv $5 \times 5$ | 32 | ReLU | Stride: 1 |
| Position Embedding | - | - | See Section A.2.4 |
| Flatten | axis: [0, $1 \times 2$, 3] | - | Flatten x, y pos. |
| Layer Norm | - | - | - |
| MLP (per location) | 32 | ReLU | - |
| MLP (per location) | 32 | - | - |

Table 6: Competition encoder layers for Tetrominoes.

| Type | Spatial Resolution | Size/Channels | Activation | Stride | Padding |
|---|---|---|---|---|---|
| Conv $5 \times 5$ | 35x35 | 32 | LeakyReLU | 1 | 2 |
| MaxPool | 17x17 | 32 | | 2 | 0 |
| Conv $5 \times 5$ | 17x17 | 32 | LeakyReLU | 1 | 2 |
| MaxPool | 8x8 | 32 | | 2 | 0 |
| Conv $5 \times 5$ | 8x8 | 32 | LeakyReLU | 1 | 2 |
| MaxPool | 4x4 | 32 | | 2 | 0 |
| Conv $5 \times 5$ | 4x4 | 32 | LeakyReLU | 1 | 2 |
| MaxPool | 2x2 | 32 | | 2 | 0 |
| Flatten | 2x2 $\rightarrow$ 4 | axis: [0, $1 \times 2$, 3] | | | |
| FC | 4 | 32 | LeakyReLU | | |
| FC | 4 | 32 | | | |

## A.3 LEARNED QUERY VISUALIZATIONS

Similarly to the Tetrominoes, we see that the recognition of objects gradually evolves for CLEVR6 and Multi-dSprites. The learning process over time of CLEVR6 is depicted in figure 9 for the reconstructions and in figure 10 for the attention heatmaps.

The learning process over time of Multi-dSprites is depicted in figure 11 for the reconstructions and figure 12 for the attention heatmaps. Although some slots do not show a visible attention map, we checked their numerical value and conclude that the attention values are chosen large enough to reconstruct the objects.

Table 7: Competition encoder layers for Multi-dSprites.

| Type | Spatial Resolution | Size/Channels | Activation | Stride | Padding |
|------|-------------------|---------------|------------|--------|---------|
| Conv 5 × 5 | 64x64 | 64 | LeakyReLU | 1 | 2 |
| MaxPool | 64x64 | 64 | | 2 | 0 |
| Conv 5 × 5 | 32x32 | 64 | LeakyReLU | 1 | 2 |
| MaxPool | 32x32 | 64 | | 2 | 0 |
| Conv 5 × 5 | 16x16 | 64 | LeakyReLU | 1 | 2 |
| MaxPool | 16x16 | 64 | | 2 | 0 |
| Conv 5 × 5 | 8x8 | 64 | LeakyReLU | 1 | 2 |
| MaxPool | 8x8 | 64 | | 2 | 0 |
| Conv 5 × 5 | 4x4 | 64 | LeakyReLU | 1 | 2 |
| MaxPool | 4x4 | 64 | | 1 | 0 |
| Flatten | 3x3 → 9 | axis: [0, 1 × 2, 3] | | | |
| FC | 9 | 64 | LeakyReLU | | |
| FC | 9 | 64 | | | |

Table 8: Competition encoder layers for CLEVR6.

| Type | Spatial Resolution | Size/Channels | Activation | Stride | Padding |
|------|-------------------|---------------|------------|--------|---------|
| Conv 5 × 5 | 128x128 | 64 | LeakyReLU | 1 | 2 |
| MaxPool | 128x128 | 64 | | 2 | 0 |
| Conv 5 × 5 | 64x64 | 64 | LeakyReLU | 1 | 2 |
| MaxPool | 64x64 | 64 | | 2 | 0 |
| Conv 5 × 5 | 32x32 | 64 | LeakyReLU | 1 | 2 |
| MaxPool | 32x32 | 64 | | 2 | 0 |
| Conv 5 × 5 | 16x16 | 64 | LeakyReLU | 1 | 2 |
| MaxPool | 16x16 | 64 | | 2 | 0 |
| Conv 5 × 5 | 8x8 | 64 | LeakyReLU | 1 | 2 |
| MaxPool | 8x8 | 64 | | 2 | 0 |
| Conv 5 × 5 | 4x4 | 64 | LeakyReLU | 1 | 2 |
| MaxPool | 4x4 | 64 | | 1 | 0 |
| Flatten | 3x3 → 9 | axis: [0, 1 × 2, 3] | | | |
| FC | 9 | 64 | LeakyReLU | | |
| FC | 9 | 64 | | | |

Table 9: Decoder layer description for CLEVR6.

| Type | Spatial res. | Size/Channels | Activation | Comment |
|------|-------------|---------------|------------|---------|
| Spatial Broadcast | 1 | 64 | - | - |
| Position Embedding | 8x8 | 64 | - | See A.2.4 |
| Conv 5 x 5 | 8x8 | 64 | ReLU | Stride: 2 |
| Conv 5 x 5 | 16x16 | 64 | ReLU | Stride: 2 |
| Conv 5 x 5 | 32x32 | 64 | ReLU | Stride: 2 |
| Conv 5 x 5 | 64x64 | 64 | ReLU | Stride: 2 |
| Conv 5 x 5 | 128x128 | 64 | ReLU | Stride: 1 |
| Conv 3 x 3 | 128x128 | 4 | - | Stride: 1 |
| Split Channels | 128x128 | RGB (3), mask (1) | Softmax on masks | - |
| Recombine Slots | 128x128 | - | - | - |

Table 10: Decoder layer description for Multi-dSprites and Tetrominoes.

| Type | Spatial res. | Size/Channels | Activation | Comment |
|---|---|---|---|---|
| Spatial Broadcast | 1 | 32 | - | - |
| Position Embedding | WxH | 32 | - | See A.2.4 |
| Conv 5 x 5 | WxH | 32 | ReLU | Stride: 1 |
| Conv 5 x 5 | WxH | 32 | ReLU | Stride: 1 |
| Conv 5 x 5 | WxH | 32 | ReLU | Stride: 1 |
| Conv 3 x 3 | WxH | 4 | - | Stride: 1 |
| Split Channels | WxH | RGB (3), mask (1) | Softmax on masks | - |
| Recombine Slots | WxH | - | - | - |

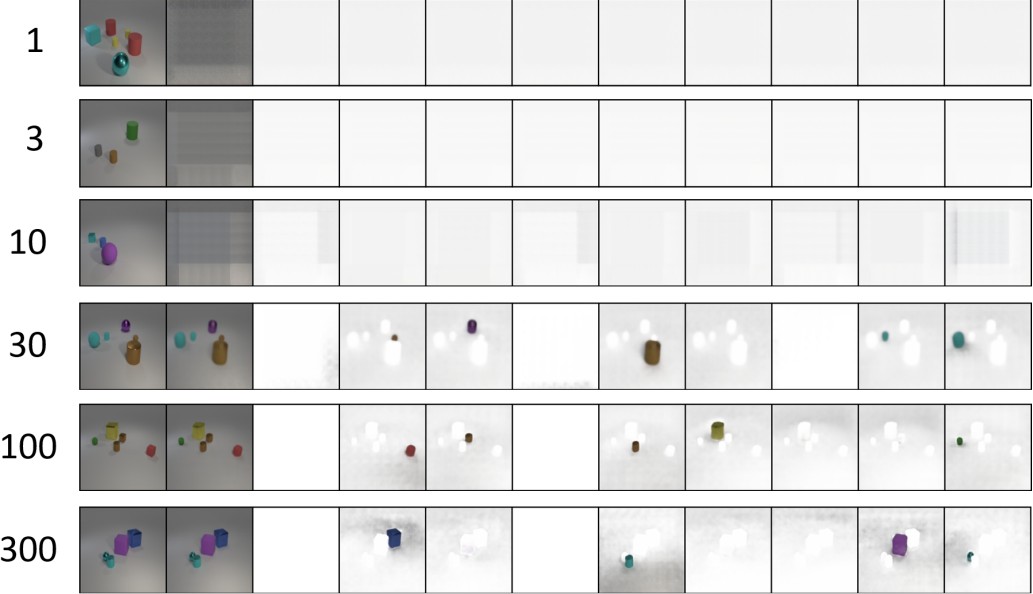

Figure 9: *Reconstructions over training of CLEVR6*: The numbers on the left denote the completed training epochs, whereas the images are reconstructions. The columns of the reconstruction images are in the following order: (col. 1) ground truth image, (col. 2) reconstruction (cols. 3-11) individual slot reconstructions.

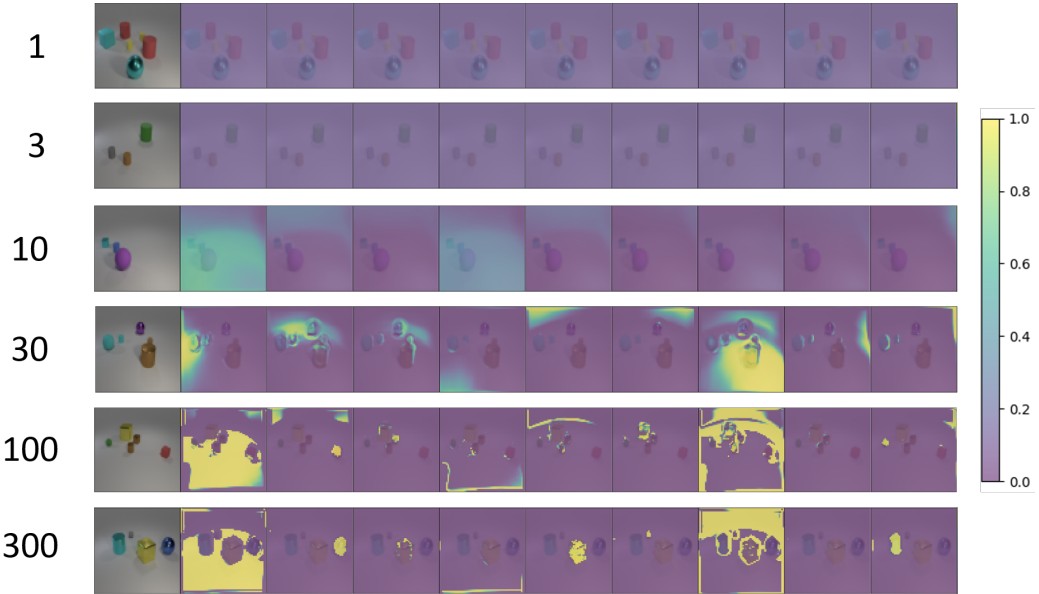

Figure 10: *Attention heatmaps during training of CLEVR6*: The numbers on the left denote the completed training epochs, whereas the images are visualized attention maps. The columns of the visualized attention maps are in the following order: (col.1) ground truth image, (cols. 2-10) individual attention maps of the queries over the keys (transformed pixel features).

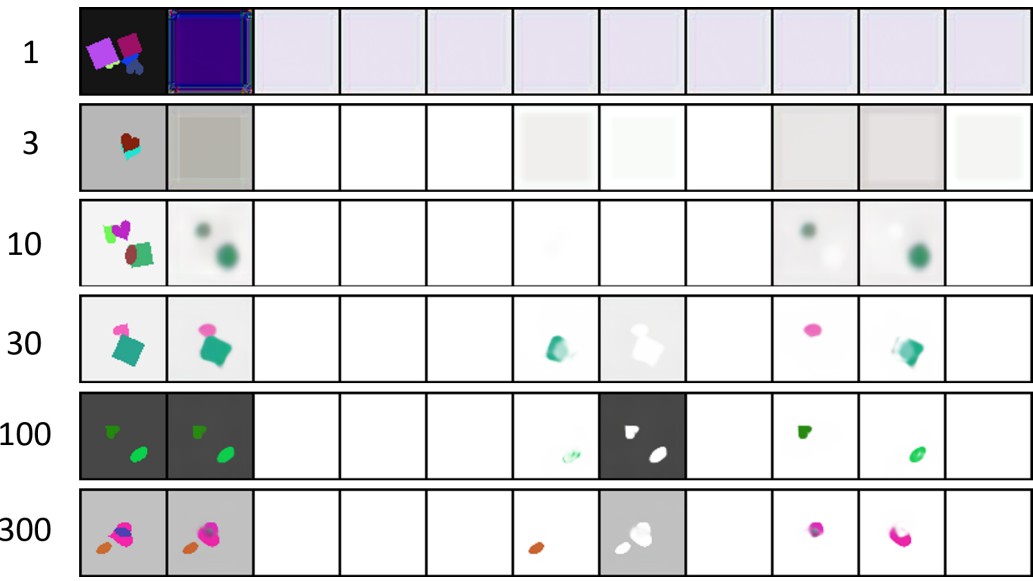

Figure 11: *Reconstructions over training of Multi-dSprites*: The numbers on the left denote the completed training epochs, whereas the images are reconstructions. The columns of the reconstruction images are in the following order: (col. 1) ground truth image, (col. 2) reconstruction (cols. 3-11) individual slot reconstructions.

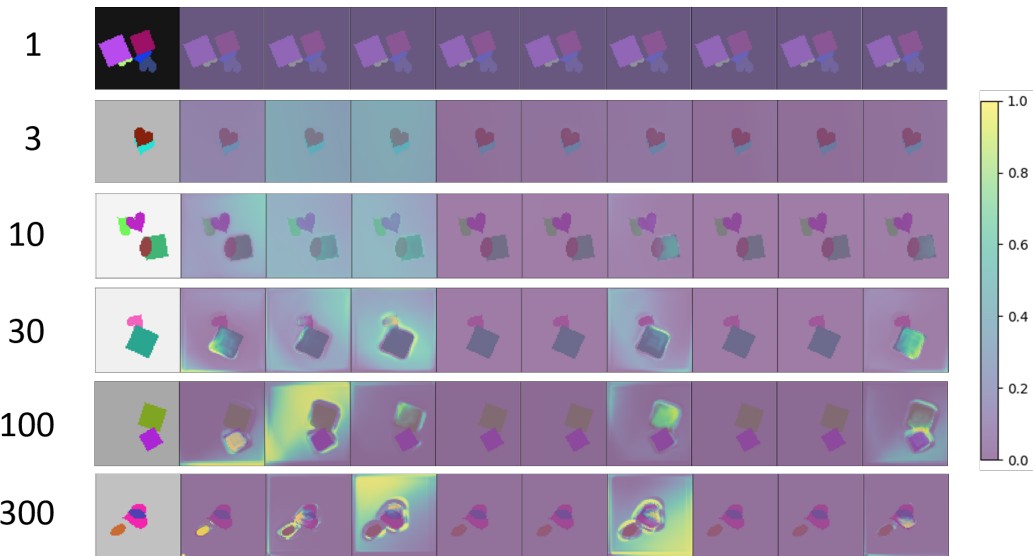

Figure 12: *Heatmaps of attention over training of Multi-dSprites*: The numbers on the left denote the completed training epochs, whereas the images are visualized attention maps. The columns of the visualized attention maps are in the following order: (col.1) ground truth image, (cols. 2-10) individual attention maps of the queries over the keys (transformed pixel features).

