# OpenReview forum: "Competition Priors for Object-Centric Learning"
_ICLR.cc/2024/Conference — Submitted to ICLR 2024_

### Official Review · Reviewer_qRty · 2023-10-14

**Soundness:** 2 fair
**Presentation:** 3 good
**Contribution:** 1 poor
**Rating:** 3
**Confidence:** 4

**Summary:**

This paper proposes COP, an object-centric learning (OCL) framework that does not involve interactive inference. COP differs from Slot Attention in that it runs Conv+Pooling layers to generate initial slots from the image feature maps. Experimental results on three simple datasets show promising results of COP.

**Strengths:**

- The non-iterative design leads to small memory consumption and runtime
- Results on simple datasets are good

**Weaknesses:**

### Novelty
- All components of COP are not new. The non-iterative attention operation is common. For example in SAVi, they also perform single iteration Slot Attention. The primitive slots come from image features, which are conceptually similar to the conditional initialization in SAVi, with the only difference that they are not from initial frame hints
- Is iterative attention a feature or a bug? To me, iterative Slot Attention enables OCL models to decompose complex data as seen in follow-up works such as STEVE [1] and LSD [2]. It is unclear to me if the non-iterative SWTA Attention can handle more complicated images (see Experiments below)

[1] Singh, Gautam, Yi-Fu Wu, and Sungjin Ahn. "Simple unsupervised object-centric learning for complex and naturalistic videos." NeurIPS, 2022.

[2] Jiang, Jindong, et al. "Object-centric slot diffusion." arXiv preprint arXiv:2303.10834 (2023).

### Experiments
- The datasets used in this paper are too simple. OCL has witnessed tremendous progress in recent years, where OCL methods have proven effective on more complex datasets [3]. The simple images tested in this paper make it hard to assess the capacity of COP. A necessary experiment is to incorporate COP with better OCL models such as STEVE, and test it on complex datasets like MOVi
- The experiments in the paper mainly focus on object segmentation. While it is an important outcome of OCL, the quality of learned object slots is another important aspect. I would suggest the authors to at least perform an object property prediction experiment on CLEVR following the protocol of Slot Attention

[3] Seitzer, Maximilian, et al. "Bridging the gap to real-world object-centric learning." ICLR, 2022.

**Questions:**

Apart from questions in Weaknesses, I have a few minor questions:
- Table 1 shows the theoretical runtime/memory complexities of COP and Slot Attention. I wonder what is the actual runtime comparison. For example, does COP improve the training speed by around the number of Slot Attention iterations?
- I do not really understand why the SWTA Attention is (significantly) better than the Slot Attention as shown in the ablation study. To me, Slot Attention applies Softmax over the slot dimension, which is a more natural way to induce slot competition

---

> ### Author Response · Authors · 2023-11-23
> **Response**
>
> We thank the reviewer for their feedback and glad that the reviewer appreciates the non-iterative design and the results on the small datasets.
>
> **Regarding SAVI:**
> Yes, SAVI is non-iterative. But, a lot of “initial” features in SAVI are manually selected. For example, centre of mass, bounding box, segmentation etc. This makes the problem much easier.
>
> **Regarding other datasets and experiments:**
> We do not have results for these datasets currently. We intend to add them in the new version of the manuscript.

---

### Official Review · Reviewer_72DU · 2023-10-26

**Soundness:** 2 fair
**Presentation:** 3 good
**Contribution:** 2 fair
**Rating:** 5
**Confidence:** 4

**Summary:**

This paper introduces a method called Competition Over Pixels (COP) that uses convolutional, max-pool, and cross-attention layers to learn object-centric representations. The convolutional and max-pool layers first learn a set of primitive slots, which are then used as queries in a single iteration of cross-attention where the softmax is done over the queries. COP is evaluated on several standard object-centric learning benchmarks, matching or outperforming several baselines in terms of FG-ARI.

**Strengths:**

The paper is generally well written and easy to understand. Using max-pooling as a mechanism to learn object-centric representations has not been done before, as far as I know. Their method also does not require multiple iterations of cross attention, which may speed up the training and inference time of this method compared to iterative refinement methods. I am encouraged by the result that COP w/o Attention can still perform well on the Tetronminoes dataset, although I would have liked to see more experiments in this direction.

**Weaknesses:**

1. The baselines used in the experiments are rather weak. The authors mention Implicit Slot Attention [1] in section 3.3 when discussing scalability, but do not compare against it in the experiments.
2. Along the same lines, it would have been informative to compare with other methods that focus on slot initialization such as a version of Slot Attention using learned slot initializations or BO-QSA [2].
3. Section 3.3 uses Implicit Slot Attention [1] as an example of a model using up to 11 iterations, but this is misleading since the experiment on number of iterations from that paper was to show that Implicit Slot Attention is robust to increasing number of iterations when compared with Vanilla Slot Attention. While it can scale to more iterations, their method does not require more iterations to perform well.
4. Since vanilla Slot Attention already does quite well on the datasets used in the paper, it would be interesting to see how COP performs on a more complex dataset such as CLEVRTex.
5. It is not clear to me from the experiments the importance of max-pooling or if some other method to learn primitive slots from the input would also work well. This is a central part of the algorithm and it would strengthen the evidence for using max-pooling if this is compared with other ways of pooling such as average-pooling or strided convolutions.
6. (minor) The original Slot Attention does not set the temperature to $\sqrt{n}$, but instead it uses the sqrt of the slot dimension.

[1] Object Representations as Fixed Points: Training Iterative Refinement Algorithms with Implicit Differentiation. https://arxiv.org/abs/2207.00787

[2] Improving Object-centric Learning with Query Optimization. https://arxiv.org/abs/2210.08990

**Questions:**

1. Do the results improve if we run multiple iterations of SWTA-Attention or Slot Attention after obtaining the primitive slots?
2. The result that COP w/o any cross attention on Tetrominoes is very interesting. Have you run this setting for any of the other datasets? Is there any noticeable difference in the segmentation masks for this setting?
3. Have you noticed significant wall-clock time difference between COP and Slot Attention?
4. What is L_comp in Table 1?
5. For the "COP w/ Attn as in Slot Attention" ablation, is the main difference the addition of the weight normalization after the softmax?

---

> ### Author Response · Authors · 2023-11-23
> **Response**
>
> We thank the reviewer for your comments and glad that you found our novel.
>
> **Regarding Baselines and Experiments:**
> We will be updating the experiments (Clevertex, multiple iterations of SWTA, BO-QSA) and baselines (ICA) in the next version of the manuscript.
>
> **Regarding Average Pooling:**
> In our experiments replacing max pooling with average pooling lower the performance considerably.
>
> **Regarding Wall Clock Time:**
> For the tetro dataset (where we have most of our experiments), we had results in half the walk clock time as Slot Attention.
>
> **Regarding "COP w/ Attn as in Slot Attention":**
> Yes, in this case we add the weight normalisation after the softmax.

---

### Official Review · Reviewer_pbAn · 2023-10-31

**Soundness:** 2 fair
**Presentation:** 2 fair
**Contribution:** 1 poor
**Rating:** 1
**Confidence:** 4

**Summary:**

The paper presents COP, a method for unsupervised object-centric learning that builds on top of Slot Attention by replacing some of its components.
In particular, the queries are not initialized at random but obtained from the features by means of max pooling, and the iterative attention is replaced with a single attention layer.
The paper motivates these choices from the perspective of "competition" and evaluates the method on standard benchmarks for object-centric learning.

**Strengths:**

I appreciate the detailed description of the encoder and decoder architectures in the appendix, as well as the number of figures where the output of the model is visualized.

**Weaknesses:**

Section 3.1 is badly structured: if the encoder and the decoder are exactly the same as Slot Attention please just refer to the original paper to save space and avoid confusion. Then, use the extra space to explain better the novelty in the object-centric bottleneck (see paragraph below).

The mechanism of Soft-Winner-Takes-All (SWTA) is not explained clearly and the comparison with Slot Attention is misleading. Slot attention takes the softmax over the queries first and then normalizes over the keys by dividing each row by its sum. As I interpret the paper, SWTA seem to perform the first operation, i.e. the softmax over the queries, but skips the normalization. Therefore, what is the novelty? Skipping the normalization? Running a single iteration instead of multiple ones? In both Slot Attention and SWTA there is competition for the patches between queries: if a patch is assigned to one query/slot it can not be assigned to others.

I strongly disagree with the analysis performed in section 3.2 about the "competition through MaxPool layers". In particular, the discussion about "sub-networks" in the following sentences: "This results in sub-networks competing to have higher activations. During back-propagation, units that win and the subnetworks that are responsible for this will get updated. As a result, a winning sub-network is reinforced to win more if it predicts correctly." I'd like to point out the CNN layers before MaxPool share the same weights across locations therefore there are no sub-networks in competition, only one network. I would like the authors to provide references to the "sub-networks" interpretation to support their claim in the paper.

In the same paragraph, it is said that "A neuron has a higher chance of winning if it explains a different part of the input, rather than explaining the same feature as another neuron." I argue that the final "primitive slots" are diverse simply because they are pooled from different regions of the input image by means of local CNN filters and local max pooling as shown in Figure 2. This is a well-known property of CNNs and it also introduces an implicit bias on the shape and size of the objects which the authors do not discuss.

Other than the query generation process the discussion about competition in section 3.1 does not add much to the original Slot Attention paper: SWTA seems to be a slight variation of their competitive attention implementation and the decoder is identical. If other papers already discussed the importance of competition in those two mechanisms, why repeating the same arguments here?

The number of slots is hardcoded and manually tuned for each dataset. This is a limitation of this method that doesn't make it stand out compared to previous ones. If anything, creating the "primitive slots" by means of max pooling is even more limiting than the sampling mechanism of Slot Attention.

Experiment design:
- The experiments only include very basic datasets where even basic color-based clustering methods would suffice. Since a few years, the object-centric community has moved on to more challenging datasets such as ClevrTex and MultiShapeNet, where the proposed method would be more interesting to evaluate.
- Also, I disagree with the choice of evaluating only the FG-ARI metric, while segmentation is a good proxy for object-centric learning, it does not capture the full complexity of the task. Other works try to evaluate the quality of the slots by means of downstream tasks such as object tracking or property prediction.
- Finally, the ablation study is not very informative. The authors should have compared with vanilla Slot Attention where 1) the query generation step is replaced with their max-pooling-based method, and 2) the iterative attention is replaced with a single attention layer, while keeping the rest unaltered. This would have provided a fair comparison and a better understanding of the contribution.

The abstract claims "scalability" as a feature of the proposed method, which I can not agree with. See the related question below.

Public code: the code in the attached zip archive is broken. For example, `train.py` tries to call a non-existing function `read_cli_args_and_get_config`. This way, it's not even possible to check that the code reflects the method described in the paper because there are multiple implementations in the zip archive. I appreciate the effort of including the code in the submission, but it should be double-checked beforehand and possibly come with some instructions.

Overall manuscript quality: many sentences are poorly written and the text would require a thorough revision. A few examples:
- "Similar to the encoder, we build the CNN layers in such a way that they preserve the spatial dimension. But, the MaxPool layers reduce it."
- "Primitive slots" are capitalized in the figure captions but never in the text. Sometimes there is a dash between the two words, sometimes not. Also it's unclear whether "primitive slots" are a new concept and how they differ from the usual term "queries", the first mention in section 3.1 could be more explicit.

**Questions:**

What is the exact formulation of SWTA? A mathematical definition, an algorithm listing, or a code snippet would be helpful. Even better, a side-by-side comparison with Slot Attention would be great.

What is the difference between the ablation study number 1 and a simple autoencoder with a resolution bottleneck?

The abstract and the intro claim that COP is a "simple, scalable, non-iterative and fully-differentiable approach". While I agree on most adjectives, I can't agree on scalability and I don't think the arguments at the end of Section 3.3 are valid. While it's true that the proposed method performs grouping in a single step, scaling to larger and more complex images will require adjusting the number of slots and therefore the computation in the stack of CNN+MaxPool layers.

**Details Of Ethics Concerns:**

The zip file attached as supplementary material contains the full `.git` folder with the commit history and the names of the authors. I found out only after finalizing the review so this knowledge had no impact on my comments.

---

> ### Author Response · Authors · 2023-11-23
> **Response**
>
> We thank the reviewer for their feedback. We will be using your comments to improve the manuscript further.
>
> **Regarding Novelty:**
>
> **SWTA vs Slot Attention:**
> Yes, you are right. SWTA-Attention skips the normalisation part after the softmax.
> But we also remove the following parts from the Slot-Attention module:
> a) Gated Recurrent Unit
> b) Skip Connection
> c) We use the same projection weights for Keys and Values. Thus reducing the number of parameters.
> d) Also, remove the iterations, which are necessary for Slot-Attention to work.
>
> This makes our method very simple, it is just, as you said, “Softmax over the queries”. Our novelty lies in showing that many of these design choices made in Slot-Attention are not necessary.
>
> **Regarding Competing Sub-Networks:**
> We have alternating CNN and Max Pool layers. At the end of these layers, a primitive slot is obtained by taking features across the “channel”. We are implying a competition between sub-networks across filters in a CNN layer, not in the shared weights. We will be improving our writing in the next version and hopefully explain it better.

---

### Official Review · Reviewer_Ep3S · 2023-11-06

**Soundness:** 3 good
**Presentation:** 3 good
**Contribution:** 3 good
**Rating:** 6
**Confidence:** 4

**Summary:**

COP is proposed as a novel method for learning object-centric representations. COP is a simple baseline for abstraction models, using vanilla building blocks like CNN, MaxPool layers, and a modified Cross-Attention. COP is evaluated on standard Object-Centric benchmarks and shows competitive or superior performance compared to other slot attention methods.

**Strengths:**

1. Performance: Our COP model surpasses the commonly adopted Slot Attention method on the Multi-dSprites and Tetrominoes datasets, setting a new benchmark in performance as measured across all three datasets.
2. Efficiency: COP demonstrates improved efficiency, boasting superior time and space complexity metrics.
3. Simplicity: The COP framework operates non-iteratively, streamlining the processing pipeline.

**Weaknesses:**

1. Scope of Dataset Evaluation: The empirical validation of the COP model is confined to synthetic datasets. While the authors posit that COP's attributes render it highly scalable for larger datasets, the absence of real-world dataset assessments renders the claims of its object-centric representation less compelling.
2. Contemporaneity of Baselines: The baselines utilized in the study are somewhat dated. A comparative analysis with more recent methodologies would be beneficial for a comprehensive evaluation, and such comparisons should be reflected in the results table.

**Questions:**

see weakness

---

> ### Author Response · Authors · 2023-11-23
> **Response**
>
> We thank the reviewer for the feedback on our manuscript. We are glad that the reviewer finds the method simple and efficient.
>
> **Regarding real world datasets:**
> We are currently working towards testing our method on real world datasets and will include the results in the updated manuscript.
>
>
> **Regarding Baselines:**
> We will also be updating the baselines we compare against. Thank you for the feedback.

---

### Meta-Review · Area_Chair_7NYC · 2023-12-04

**Metareview:**

The method presents COP (Competition over pixel features), a method for unsupervised object segmentation. Specifically, it builds on Slot Attention and extends/simplifies it in terms of slot initialization and grouping.

While the proposed method is simple and efficient, all the reviewers agree that this paper currently does not meet the bar for acceptance in terms of experiment design/evaluation (e.g. baselines), validation of scientific claims, and has insufficient scope in terms of dataset complexity/applicability, given that the field has significantly advanced since the publication of the Slot Attention paper (which this paper is primarily based on). Given the elegance and simplicity of the proposed method, I recommend validating the approach on a wider set of datasets and tasks against more recent baselines when revising the paper.

**Justification For Why Not Higher Score:**

Insufficient experimental validation.

**Justification For Why Not Lower Score:**

N/A

---

### Decision · Program_Chairs · 2024-01-16

Reject